# The Role of vIL-6 in KSHV-Mediated Immune Evasion and Tumorigenesis

**DOI:** 10.3390/v16121900

**Published:** 2024-12-10

**Authors:** Somayeh Komaki, Tomoki Inagaki, Ashish Kumar, Yoshihiro Izumiya

**Affiliations:** 1Department of Dermatology, School of Medicine, University of California Davis, Sacramento, CA 95817, USA; 2Department of Biochemistry and Molecular Medicine, School of Medicine, University of California Davis, Sacramento, CA 95817, USA

**Keywords:** vIL-6, KSHV, transcription, immune response, oncogenesis

## Abstract

Kaposi’s sarcoma-associated herpesvirus (KSHV) is a double-stranded DNA gamma herpesvirus. Like other herpesviruses, KSHV establishes a latent infection with limited gene expression, while KSHV occasionally undergoes the lytic replication phase, which produces KSHV progenies and infects neighboring cells. KSHV genome encodes 80+ open reading frames. One of the KSHV genes, K2, encodes viral interleukin 6 (vIL-6), a homolog of human IL-6 (hIL-6), mainly expressed in the lytic phase of the virus. vIL-6 plays a crucial role in regulating the expression of other viral genes and is also associated with inducing angiogenesis, cell survival, and immune evasion, which is suggested to promote the development of KSHV-associated diseases. This review summarizes the current knowledge on vIL-6. We focus on the vIL-6 regarding its protein structure, transcriptional regulation, cell signaling pathways, and contribution to the KSHV-associated diseases.

## 1. Introduction

Cytokines are signaling proteins that stimulate host immune response and control homeostasis in our body. They include a family of interleukins (ILs), interferons (IFNs), tumor necrosis factor (TNF), and other chemokines. For activating cell signaling, each cytokine binds to a specific receptor on its target cells [1]. They are normally secreted from cells for intercellular signaling, while some function within the cell by binding to the intracellular receptor. Since they regulate multiple cellular responses such as inflammation, growth, and maturation, their dysregulation is associated with various disease development [2,3].

Viral infection and subsequent host response are strongly associated with the disease development, in unfortunate cases, leading to uncontrolled cytokine production. Kaposi’s sarcoma-associated herpesvirus (KSHV) is one of the oncogenic gamma herpesviruses and encodes a viral cytokine; viral IL-6 (vIL-6) [4,5]. vIL-6 is a homolog of human IL-6 (hIL-6), in structure, receptor binding, and biological functions [6,7,8]. The evolutionary origins of vIL-6 can be traced back to the adaptation of KSHV to manipulate host immune responses for its survival. Other mammalian herpesviruses, such as herpes simplex virus (HSV) and Epstein–Barr virus (EBV), do not encode an IL-6 homolog [9]. KSHV has developed unique adaptations during evolution, allowing the virus to stimulate hIL-6 signaling pathways more effectively. vIL-6 has evolved to mimic hIL-6’s function [10]. The vIL-6 is highly expressed in KSHV-replicating cells and is also expressed in a small population of latently infected cells at low concentrations [4,11]. The experiments using cell lines with minimal background reactivation strongly suggest that vIL-6 may be expressed independently of full lytic activation. However, the possibility of spontaneous reactivation in a small subset of cells cannot be entirely ruled out. The detection of latent vIL-6 transcripts and its variability across cell lines highlight KSHV’s flexible transcription program, enabling the virus to adapt its gene expression to distinct cellular environments or stress conditions [11]. In a murine xenograft model, the vIL-6 transgenic mice developed a multicentric Castleman disease (MCD)-like disease and supported tumor metastasis [12]. vIL-6 is clinically detectable in the blood and tissues of patients with KSHV-associated diseases such as Kaposi sarcoma (KS), primary effusion lymphoma (PEL), and MCD [13]. KSHV inflammatory cytokine syndrome (KICS) is also characterized by a high KSHV viral load and sustained elevations of serum vIL-6, which lead to severe systemic inflammation [14,15]. Because of abundant expression and the ability to induce inflammatory signaling in infected cells and surrounding tissues, vIL-6 plays a critical role in developing KSHV-associated diseases. Since the discovery of KSHV in 1994 [16], KSHV research communities as a whole uncovered multiple biological functions of vIL-6. vIL-6 may also play different roles depending on whether the virus is in the latent or lytic phase. In the latent phase, small amounts of vIL-6 [11] can activate downstream cellular signaling, which might contribute to cell survival and proliferation through STAT3 signaling activation. On the other hand, during the lytic phase, where infected cells die, and vIL-6 is released extracellularly, vIL-6 might contribute to viral replications rather than cell survival. Here, we will focus on the vIL-6 localization and viral transcription regulation, receptor engagement, associated cell signaling pathways, and association in KSHV-related diseases.

## 2. Localization and Transcription Regulation of vIL-6

KSHV, also known as human herpesvirus 8 (HHV-8), was initially detected in KS lesions in 1994 [16]. KSHV is an oncogenic gamma herpesvirus [17] with an enveloped virion and a double-stranded DNA genome [18]. The KSHV genome consists of a central coding area of roughly 145 kb of DNA, flanked by approximately 30 kb of terminal repeats (TRs) [19,20]. KSHV establishes a latent infection with limited gene expression [21] that lasts throughout the host’s life [22]. Upon reactivation, KSHV expresses viral genes, including K2, encoding vIL-6 to efficiently produce progeny viruses. In this section, we focus on the localization and transcription regulation of vIL-6.

(i)Localization

vIL-6 becomes detectable within 10 h after reactivation and increases for another 2 days [21]. Although vIL-6 is known to be expressed at low concentrations in latently infected cells, vIL-6 has been categorized as the primary early lytic gene, since the expression of vIL-6 is explosively increased during viral replication at early time points [11,23,24]. A significant amount of vIL-6 remains within the ER, resulting in less efficient secretion compared to hIL-6 [25,26]. The endoplasmic reticulum (ER) chaperone protein, calnexin, affects the localization and cellular retention of vIL-6 [27]. In addition, vIL-6 can bind to the ER-localized protein, the nonsignaling membrane receptor vitamin K epoxide reductase complex subunit 1 variant 2 (VKORC1v2) [28], thereby inhibiting the viability of PEL cells and the replication of the progeny virus [28].

The receptor for the IL-6 family consists of an alpha subunit (IL-6R) and a beta subunit (gp130) [5]. Blocking the interaction between vIL-6 and gp130, specifically in the ER, by using the gp130 dimerization-defective vIL-6 variant (W167G) reduces cell proliferation and viability [29], highlighting the importance of the vIL-6 ER localization. Importantly, the exogenous supplementation of the culture supernatant with vIL-6 did not restore the production of the vIL-6 knockout virus [30]. Following lytic reactivation, the death of KSHV-infected cells can result in the release of vIL-6 into surrounding tissues [31]. The differences in vIL-6 primary localization at different KSHV replication stages and their associated biological roles could be subjects for future investigation.

(ii)Transcription Regulation

The KSHV lytic phase is triggered by the activation of the open reading frame (ORF) 50, known as a replication and transcription activator (Rta). The Rta activates vIL-6 through direct or indirect interaction with the vIL-6 promoter by triggering Notch signaling, mediated by the recombination signal-binding protein for Ig kappa J region (RBP-Jκ) [32,33,34]. vIL-6 expression is also increased by ORF57, known as mRNA transcript accumulation (Mta). The coding region of vIL-6 contains an Mta-responsive element (MRE) consisting of two specific binding sites, MRE-A and MRE-B. Mta binds to MRE-A and stabilizes vIL-6 RNA. In the cytoplasm, Mta binds MRE-B, which inhibits the binding of the miRNA-containing RNA-induced silencing (RISC) complex (miR-1293) to the vIL-6 mRNA [19,35]. It prevents the formation of stress granules by attaching to protein kinase R (PKR), blocking PKR activation and the phosphorylation of eukaryotic translation initiation factor 2 (eIF2α) [19,35].

In addition to Mta, the glycoproteins B (gB) and K8.1, which function in virion attachment to cell surface integrins and virus entry as late lytic genes, also positively regulate the expression of vIL-6. The treatment of PEL cells with neutralizing antibodies against the gB and K8.1 leads to a notable decrease in the vIL-6 and vascular endothelial growth factor (VEGF), inhibiting angiogenesis [13]. In the contexts of iSLK and iTIME cells, vIL-6 expression is inhibited by viral interferon regulatory factor-1 (vIRF-1) through its regulation of tyrosine kinase (TYK2), which may involve the suppression of interferon (IFN)-I signaling. However, the indirect effects of vIRF-1 might mediate the positive regulation of the viral cytokine, contributing to the balanced expression of vIL-6 [36,37]. The levels of vIL-6 were significantly reduced in BCBL-1/anti-K9 (vIRF-1) cells compared to those observed in BCBL-1 cells [38]. Hypoxia plays an important role in the oncogenesis of KSHV-induced tumors, with hypoxia-inducible factors (HIFs) serving as the primary mediators of the cellular response. In addition, even under normal oxygen conditions, HIF-1 is crucial for expressing KSHV lytic genes in PEL cell lines. HIF-1α knockdown resulted in a decrease in vIL-6 expression as well as viral production [39].

The transcription factor X-box binding protein-1 (XBP-1s), activated by ER stress, can directly bind to the XBP-response elements (XREs) in the promoter region of vIL-6 and initiate its activation in latently infected KSHV cells [40]. When BCBL-1 cells were treated with tunicamycin, a chemical that induces XBP-1s, there was an upregulation of vIL-6 [40]. XBP-1s can also stimulate human IL-6 expression, creating a positive feedback loop that further enhances the effects of vIL-6 and supports the survival of infected cells [41,42]. Importantly, vIL-6 positively regulates the expression of lytic genes, including Rta and viral DNA replication-related genes [10,30].

## 3. Receptor Engagement of vIL-6 and hIL-6

vIL-6 shares structural similarities, receptor utilization, and biological activity with hIL-6. The IL-6 homolog is absent in other herpesviruses, such as herpes simplex virus and Epstein–Barr virus (EBV) [9]. While hIL-6 has 212 amino acids, vIL-6 comprises 204 amino acids with 24.8% of sequence homology and 62.2% of similarity in their amino acid composition [4,43,44]. While the molecular weight of vIL-6 is approximately 22.6–24 kDa, hIL-6 is approximately 23.7 kDa [45,46]. Despite their structural similarity, vIL-6 exhibits a signaling potency that is 100 to 1000 times less than hIL-6 in vitro [6]. This weakened activity likely may help KSHV avoid excessive inflammatory responses during the latent phase. Pulse-chase analysis showed that the half-time of the recombinant vIL-6 secretion is approximately 4 h, 8 times longer than hIL-6 [47]. vIL-6 can substitute the function of hIL-6 and maintain the viability of human myeloma cell lines [43,45,48,49,50]. Table 1 shows the basic similarities and differences between vIL-6 and hIL-6. In this section, we focus on the differences between vIL-6 and hIL-6 in terms of their receptor engagement.

(i)Cellular tropism of vIL-6 and hIL-6

The receptor for the hIL-6 cytokine family consists of an hIL-6 receptor alpha subunit (IL-6R), which binds to hIL-6 strongly, and a beta subunit known as gp130. This receptor is utilized in common with hIL-6 family cytokines, including oncostatin M, IL-11, leukemia inhibitory factor (LIF), and novel neurotrophin-1/B-cell stimulating factor-3 [5,51]. Upon hIL-6’s attachment to IL-6R, gp130 undergoes the homodimerization and phosphorylation of specific tyrosine residues, initiating several downstream signaling pathways. Cells without IL-6R are unresponsive to hIL-6, as it cannot bind effectively to gp130; this restricts signaling since IL-6R’s expression is limited to epithelial cells, hepatocytes and, certain leukocytes [12,52].

vIL-6 signals by forming tetrameric complexes. The gp130 signal transducer mediates the effects of vIL-6 and does not rely on the IL-6R component of the receptor–signal transducer complex. However, vIL-6 is also capable of binding to IL-6R, creating a hexameric complex. The hexameric complex possesses greater signaling efficacy and complex stability [25,53,54,55,56]. Since vIL-6 does not require IL-6R for downstream signaling and many types of cells in our body express gp130, vIL-6 can potentially induce more widespread effects than traditional hIL-6 signaling [4,5,47,57]. vIL-6 transgenic mice exhibit systemic symptoms like MCD, including hyperplasia of the spleen and lymph nodes, hypergammaglobulinemia, plasmacytosis in peripheral lymph nodes, and splenic extramedullary hematopoiesis [12]. The expanded tropism of vIL-6, which only requires gp130 to activate downstream signaling, may contribute to the severe systemic symptoms observed. KICS patients, who are characterized by high levels of vIL-6, experience severe systemic inflammation [15].

The differences in receptor engagement between vIL-6 and hIL-6 may represent an adaptation of KSHV, allowing vIL-6 to signal to a broader range of target cells that have downregulated IL-6R to protect against hIL-6 hyperstimulation [5].

(ii)Binding sites of vIL-6 and hIL-6

IL-6 family cytokines interact with their receptors at three sites; site I engages with the nonsignaling receptor, IL-6R. Site II engages with the D2D3 region of gp130, and site III with the D1 region of a second gp130 signaling receptor [53,58,59]. Site II and III epitopes of vIL-6 have more hydrophobic content compared to hIL-6 [53]. vIL-6 binds directly to gp130 at sites II and III, but site I, which usually binds to the IL-6R, remains unoccupied [60]. vIL-6 and hIL-6 both engage with site II [53]. The amino acid residues in vIL-6 that align with sites I and III in hIL-6 play a critical role in IL-6Rα-dependent signaling [55]. The corresponding sequences of hIL-6 can replace the N- and C-terminal parts of vIL-6 without losing the signal ability in an IL-6R-independent manner [61]. However, if other regions of vIL-6, such as helix A, B, C, the first half of D (not the distal region of helix-D), and the A/B loop are swapped with their counterparts in hIL-6, this IL-6R independent signaling is lost (Figure 1A) [61].

Using neutralizing anti-vIL-6 monoclonal antibodies (mAbs), which specifically bind to a domain within site I (on the C-terminal of the A/B loop and the start of the B helix of vIL-6), showed that this binding may affect the conformation of sites II and III. In hIL-6, the corresponding region interacts with IL-6R. Additionally, these mAbs hindered vIL-6’s binding to soluble gp130. The mAbs bindings were mapped outside the binding surface to gp130, suggesting that these mAbs might have blocked necessary conformational changes for vIL-6 binding to gp130. This may also explain the vIL-6 having a 1000 times weaker binding affinity to gp130 compared to the hIL-6/IL-6R complex’s affinity [58].

(iii)Glycosylation of vIL-6 and hIL-6

Many cytokines undergo posttranslational modification by oligosaccharides, impacting protein folding and maturation, biological functions, molecular stability, receptor usage, and signaling [62,63]. The degree of glycosylation and their attached sites determine the variations in biological functions and receptor binding of hIL-6 and vIL-6 [63]. Specific glycans located at Asparagine-89 on vIL-6 protein enable vIL-6 to independently interact with gp130 without IL-6R [64]. Unlike hIL-6, these sites are fully glycosylated in vIL-6 [47]. One study demonstrated that vIL-6’s N-linked glycosylation enhances vIL-6’s ability to bind to gp130 and downstream signaling pathways [64]. In contrast, hIL-6’s N- or O-linked glycosylation is not crucial for hIL-6 binding to gp130. Unlike vIL-6, glycosylated or unglycosylated hIL-6 retains its potency in B-cell proliferation induction, regardless of glycosylation status [64].

**Table 1 viruses-16-01900-t001:** The list of biological characteristics on vIL-6 and hIL-6.

	Differences	References
** Amino Acid Similarity **	vIL-6 has 62.2% amino acid composition similarity with hIL-6.	[4,43,44]
** Sequence Homology **	vIL-6 has approximately 24.8% sequence homology with hIl-6.	[4,43,44]
** Molecular weight **	Molecular weight of vIL-6 is 22.6–24 kDa and molecular weight of hIL-6 is 23.7 kDa.	[45,46]
** Cell Expression in KS **	vIL-6 expresses in only 1% to 2% of cells but hIL-6 expresses abundantly.	[43,65]
** Secretion Half-time **	Secretion half-time of vIL-6 is 8 times longer than hIL-6.	[47]
** Signaling Potency **	Signaling potency of vIL-6 is 100 to 1000 times less than hIL-6.	[6]
** Receptor Interaction **	vIL-6 can directly bind to gp130 without needing IL-6R, forming a tetramer complex. hIL-6 requires IL-6R and gp130, forming a hexamer complex.	[25,53,54,55,56]
**Location**	vIL-6 is located mostly in ER, but hIL-6 is effectively secreted.	[25,26]
**Posttranslational modification**	vIL-6 needs glycosylation to be fully functional, hIL-6 does not need glycosylation to be functional.	[64]
**Pathways**	vIL-6 and hIL-6 have similar signaling pathways including JAK/STAT, MAPK, and PI3K/Akt pathways.	[5,49,66]
**Target cells**	vIL-6 can affect a wider variety of cells expressing gp130. hIL-6 affects cells expressing IL-R and gp130 both like hepatocytes, leukocytes, and epithelial cells.	[4,5,47,57]

## 4. Signaling Pathways of vIL-6

vIL-6 plays a crucial role in promoting tumorigenesis by driving cell survival, proliferation, and angiogenesis. It enhances angiogenesis by upregulating the vascular endothelial growth factor (VEGF) and downregulating caveolin 1 (CAV1), which promotes endothelial cell growth and vascular development [67,68,69]. Like hIL-6, vIL-6 exerts its effects through the activation of the JAK/STAT, Ras/MAPK, PI3K/Akt, and H7-sensitive pathways (Figure 1B) [5,49,66]. In this section, we will overview the representative signaling pathways of vIL-6 (Table 2).

(i)JAK/STAT pathway

The JAK/STAT signaling cascade begins with the dimerization of gp130 by catalyzing the phosphorylation of crucial tyrosine residues on gp130. This phosphorylation then activates Janus kinases (JAKs) such as JAK1, JAK2, and Tyk2 [70]. Subsequently, the signal transducer and activator of transcription 1 (STAT1) and STAT3 proteins bind to these phosphorylated tyrosine residues and undergo phosphorylation. They then dimerize and translocate to the cell nucleus, where they activate the inflammation pathway [5,71]. Similar to hIL-6, vIL-6 triggers JAK/STAT signaling by binding to gp130 [5].

Integrins are membrane glycoproteins that act as receptors for growth factors and cytokines. Integrins initiate “outside–inside” signaling upon ligand binding [72] and promote angiogenesis [73,74,75,76,77]. vIL-6 promotes an increase in the integrinβ 3 subunit (ITGB3) expression in endothelial cells by JAK/STAT pathway, contributing to its role in angiogenesis [77]. Notably, this induction of ITGB3 is unique to vIL-6, and hIL-6 overexpression does not alter ITGB3 levels [77].

vIL-6 also induces the upregulation of DNA methyltransferase 1 (DNMT1) and methylates the CAV1 promoter, leading to the downregulation of CAV1 and promoting angiogenesis via STAT3 [67,78].

vIL-6 enhances the expression of carcinoembryonic antigen-related cell adhesion molecule 1 (CEACAM1) through the JAK/STAT pathway [69]. CEACAM1, a transmembrane adhesion protein in endothelial cells, enhances the migration of endothelial cells, promoting angiogenesis, contributing to vascular remodeling, and is essential for the survival of infected B cells in PEL and MCD [79,80,81,82,83]. Phosphorylated CEACAM1 increases cell motility by decreasing the activation of focal adhesion kinase and interacting with the integrin regulator. Like ITGB3, hIL-6 does not increase the expression of CEACAM1 despite its ability to initiate the JAK/STAT pathway [69].

Hypoxia-upregulated protein 1(HYOU1), a member of the ER stress protein family, amplifies the capacity of vIL-6 to engage with gp130, subsequently boosting the JAK/STAT pathway activation [84]. HYOU1 is overexpressed under hypoxic conditions and contributes to the movement and metastasis of cancer cells in various human cancers [85].

(ii)Ras/MAPK Pathway

The mitogen-activated protein kinase (MAPK) pathway consists of a series of proteins, such as RAS, RAF, MEK, and ERK, which play a role in cell survival and proliferation [86,87]. RAF kinase is responsible for the sequential activation of downstream targets, such as MEK and the transcription factor ERK, which plays multiple roles in cellular processes, such as cell cycle, cell proliferation, and cell survival [88]. vIL-6 can activate RAS/MAPK [5,26,49,66].

(iii)PI3K/Akt Pathway

vIL-6 activates the phosphatidylinositol 3-kinase (PI3K)/protein kinase B (Akt) pathway by binding to its receptor gp130. This leads to the PI3K-mediated production of PIP3, which recruits and activates Akt, a key player in promoting cell survival, proliferation, and angiogenesis [89].

Many studies have shown that individuals infected with KSHV who are also HIV-1 positive have an increased risk of developing KS-associated diseases compared to those who are HIV-negative [90,91]. One study demonstrated that vIL-6 works synergistically with the HIV-1-secreted protein, trans-activator of transcription (Tat), to promote tumorigenesis and angiogenesis. In vIL-6-expressing epithelial cell lines with the exogenous expression of Tat, it activates PI3K and Akt pathways while deactivating tumor-suppressing proteins such as PTEN and GSK-3β. Tat upregulates the expression of VEGF, b-FGF, and cyclin D1, promoting angiogenesis [92]. Considering that Tat does not directly increase the vIL-6 expression, its impact on tumorigenesis and angiogenesis might be due to indirect effects such as the activation of cellular signals [92]. In addition to Tat, another HIV-secreted protein named negative factor (Nef) facilitates the angiogenesis and tumor development induced by vIL-6 by activating the PI3K/Akt pathway [68].

**Table 2 viruses-16-01900-t002:** Key molecules mediated by vIL-6 in tumorigenesis and angiogenesis.

Molecule	Function/Pathway	References
Integrinβ 3 subunit (ITGB3)	Promoting angiogenesis by JAK/STAT pathway activation	[77]
DNA methyltransferase 1 (DNMT1)	Downregulation of CAV1 and promoting angiogenesis via STAT3	[67,78]
Carcinoembryonic antigen-related cell adhesion molecule 1 (CEACAM1)	Migration of endothelial cells, promoting angiogenesis, via JAK/STAT pathway	[69]
Hypoxia-upregulated protein 1 (HYOU1)	Boosting vIL-6 to engage with gp130 and JAK/STAT pathway activation	[84]
Trans-activator of transcription (Tat)	Tumorigenesis and angiogenesis via activation of PI3K and Akt pathway	[92]
Negative factor (Nef)	Tumorigenesis and angiogenesis by activating the PI3K/Akt pathway	[68]

**Figure 1 viruses-16-01900-f001:**
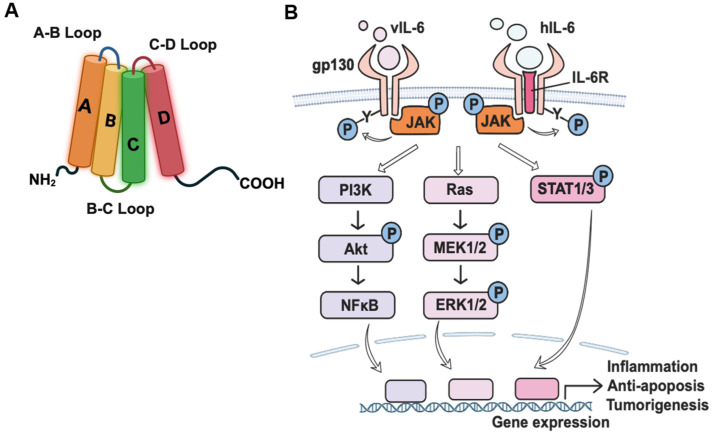
(**A**) Schematic structure model of vIL-6 (Biorender). (**B**) vIL-6 and hIL-6 signaling complexes and JAK/STAT, RAS, and PI3K signaling pathways.

## 5. Manipulation of Host Immune Response by vIL-6

The host immune response to viral infection involves innate and adaptive immune systems, which work together to detect and eliminate the virus, prevent its spread, and develop long-term immunity [93]. The innate immune system is the first line of defense and responds rapidly to viral infection within hours. The adaptive immune system is activated when the virus evades the innate response. This response is specific to the virus and takes longer to develop [94].

(i)Innate Immune Response(i-1)Inhibition of Immune Cell Infiltration by vIL-6

Cytokines act as messengers that coordinate the activities of immune cells, regulate the immune response, and help the body respond rapidly to infections before the adaptive immune response [95]. IL-1β is a pro-inflammatory cytokine secreted by macrophages and monocytes and plays a critical role in the immune response, particularly in recruiting immune cells [96,97]. vIL-6 inhibits the expression of IL-1β-induced C-X-C motif chemokine ligand 8 (CXCL8), preventing neutrophil infiltration during B-cell infection. This reflects the regulatory role of vIL-6 in leukocyte recruitment and suppressing innate immune responses, which might affect the progression of KSHV-associated disease [98]. In addition, vIL-6 can upregulate the expression of the D6 decoy receptor, which scavenges pro-inflammatory CC chemokines, keeping the surface of lymphatic endothelial cells (LECs) free of CC chemokines. This reduces the inflammatory response and the recruitment of immune cells to the tumor microenvironment [99].

(i-2)Suppressing Interferon Signaling by vIL-6

The human immune system produces interferons (IFNs) in response to viral infections to create an antiviral state in cells and trigger apoptosis and growth arrest [100]. IFNs contribute to initiating G1/S cell cycle arrest by promoting the production of the cyclin-dependent kinase inhibitor p21CIP1/WAF1 [37,101]. The promoter of vIL-6 contains two interferon-sensitive response element (ISRE) regions; ISRE-1 at the −509 to −496 bp upstream from the vIL-6 translation initiation region, and ISRE-2 is situated at −420 to −401 bp. Both elements are necessary for the interferon-induced activation of vIL-6 transcription [20,37]. In PEL cell lines, including BCP-1 and BC-1 cells, treatment with IFN-α activates vIL-6 transcription. vIL-6 then suppresses the IFN-induced p21 cyclin-dependent kinase inhibitor, creating a negative feedback loop [37,102]. Electrophoretic mobility–shift assays demonstrated that vIL-6 blocks IFN signaling by inhibiting interferon-stimulated gene factor 3 (ISGF3) binding to the ISRE probe. This inhibition is mediated by interfering with the phosphorylation of Tyk2 kinase at the IFN receptor. As a result, the JAK-STAT signaling pathway detaches, impairing the cell’s ability to respond to interferon signaling. In contrast, hIL-6 has no such effect [37]. IFN-α inhibits hIL-6-induced but not vIL-6-induced gp130 tyrosine phosphorylation mostly at the receptor level and mainly posttranscriptional. Also, treatment with IFN-α leads to the downregulation of the surface expression of IL-6R while not affecting the surface expression of gp130 [37,103].

(ii)Adaptive Immune Response(ii-2)Th2 cell Polarization and B-Cell Modulation by vIL-6

T helper2 (Th2) cells rather than T helper1(Th1) cells drive the adaptive immune response in KSHV [104,105]. vIL-6 enhances the basal and IL-1β-induced expression of the C-C Motif chemokine (CC chemokine), such as C-C Motif Chemokine Ligand 2 (CCL2), which is implicated in directing Th2 cell polarization. Th2 polarization and Th2-induced humoral immunity through B cells result in a weaker antiviral response compared to Th1-mediated immunity [98,106]. B cells, the major player in humoral immunity activated by Th2 cells, are crucial to the development and progression of KSHV-associated diseases by acting as reservoirs, facilitating the spread of the virus, and secreting cytokines [104,107,108]. During the cancer development, KSHV-infected B cells exhibit mutated immunoglobulin genes and markers like CD45, CD38, and CD138 and lack B-cell markers such as CD19 and CD20. This is consistent with the absence of B-cell-associated antigens in PEL cell lines [109,110]. PEL cells express markers that exhibit characteristics of both plasma cells and immunoblasts, corresponding to an intermediate stage in B-cell development between these two cell types [111]. In KSHV-infected BJAB cells, vIL-6 absence was associated with the pro-apoptotic marker CD30. In contrast, pro-growth markers such as CD45 increased in cells containing vIL-6, suggesting that vIL-6 promotes cell growth and inhibits apoptosis during B-cell infection [109].

B cells are able to change an antibody’s isotype by class-switching recombination (CSR), to preserve antigen specificity and enhance its effector function [112]. Activation-induced cytidine deaminase (AID) is the key mediator of antibody diversification for CSR by targeting highly repetitive switch regions to mediate DNA double-stranded breaks (DSBs) [113]. Unlike hIL-6, vIL-6 promotes class-switching recombination with an increased expression of AID in murine B cells [104]. Consistent with that observation, in vIL-6-dependent manner KSHV has altered the specificity of the immunoglobulin light chain and enhanced the switching to IgG1 and IgA isotypes [104,114]. Immunoglobulin class-switching isotypes will help KSHV evade the strong systemic immune response [114]. In addition, vIL-6 but not vIL-6 knock-out KSHV-infected monocytes showed a decrease in the expression of MHC class II genes (HLA-DR), which is vital for presenting the antigens to T cells. These monocytes showed a unique transcriptional profile indicative of immune suppression, reduced capacity for T-cell stimulation, and increased survival rate compared to those lacking vIL-6 or uninfected controls [10].

## 6. Association of vIL-6 in KSHV-Related Diseases

In individuals without immunodeficiencies, KSHV infection typically does not present clinical symptoms and remains unnoticed despite periods of lytic activation [57]. vIL-6 contributes to the pathogenesis of KS, PEL, and MCD, with its role potentially differing in these conditions due to the virus’s varying extents of lytic replication. In total, 2–5% of PEL cells and 5–25% of LANA-expressing MCD tumor cells express vIL-6 [115]. In a group of HIV-1 patients, serum vIL-6 was found in 38.2% of KS patients and 85.7% of PEL and MCD patients [116]. On average, the vIL-6 transcription in both PEL and MCD was at least ten-fold greater than the low amounts in a subset of KS [117,118]. On the other hand, no clear correlation was found between the levels of vIL-6 in the patient’s serum and the occurrence of malignancies related to KSHV [116]. Here, we will take a closer look at the association of vIL-6 with each KSHV-associated disease.

(i)Kaposi’s Sarcoma (KS)

KS is a tumor with lymphatic endothelial system origin, expressing endothelial cell markers, and thus presents as a vascular-rich tumor. It is typically found on the skin, less commonly in the oral cavity, gastrointestinal system, and lungs [119,120,121,122]. vIL-6 was first detected in KS lesions [116]. vIL-6 induces differentiation of the blood endothelial cells into the lymphatic endothelial cells (LEC), which are thought to be the origin of KS tumors, by activating the JAK/STAT and PI3k/AKT pathways [123]. The vIL-6-positive patients mostly have severe forms of KS with visceral involvement [124]. In KS lesions, only 1% to 2% of the cells express vIL-6, while those expressing hIL-6 are significantly more [43,65].

(ii)Primary Effusion Lymphoma (PEL)

PEL, typically found in immunocompromised individuals with AIDS, is a non-Hodgkin lymphoma that manifests primarily as pleura and peritoneum lymphomatous effusions, mostly without solid tumor formation [125,126]. vIL-6 was found to contribute to an increased number of tumors in an immunocompromised mouse model of B-cell lymphoma [109]. A monoclonal antibody targeting vIL-6 was shown to inhibit the proliferation of the PEL cell line and the activation of STAT3. This inhibition occurs through the antibody’s binding to the region on vIL-6 that interacts with gp130, thereby blocking its signaling function [4,127].

(iii)Multicentric Castleman Disease (MCD)

MCD is a lymphoproliferative disorder associated with waxing and waning symptoms such as fever, cachexia, and fatigue accompanied by lymphadenopathy and splenomegaly. The flare-ups, which are often accompanied by autoimmune hemolytic anemia and gammopathy, if left untreated, rapidly progress and have a poor prognosis. Although the long-term outcomes are favorable with proper treatment nowadays (71% overall survival at 10 years for 62 patients), the vIL-6-positive MCD patients have been reported to experience higher fatality rates [48,57,128,129,130]. vIL-6 has been reported to stimulate the expression of human IL-6 in cell lines derived from MCD patients [12]. MCD patients, who have higher expression levels of hIL-6 and vIL-6, show higher C-reactive protein (CRP), worse hyponatremia, higher KSHV viral load, and higher IL-10 compared to those who have higher expression levels of only hIL-6 [131]. This indicates that both vIL-6 and hIL-6 have the potential to cause severe KSHV-associated MCD symptoms. A recombinant soluble gp130Fc (sgp130) protein, a dimerized fusion protein combining sgp130, and the constant region of human IgG1, inhibits the IL-6/sIL-6R complex and vIL-6 [4,12,127,132,133]. These results suggested that vIL-6 is a promising therapeutic target for patients with MCD. Since the hIL-6 monoclonal antibody (tocilizumab) binds to IL6R, while vIL-6 binds to the gp130 without the requirement of IL6R, the effect of this treatment on KSHV patients is likely to be incomplete [134]. Although the neutralizing antibody of vIL-6 is not commercially available now, it is important to examine the contribution of vIL-6 in the pathogenesis of MCD to explore the efficacy of treatments, including combination therapy.

(iv)KSHV inflammatory cytokine syndrome (KICS)

KICS patients are characterized by a sustained elevation of vIL-6 [15]. KICS shares the pathophysiology mechanism and clinical presentation with MCD, but unlike MCD, individuals with KICS do not present with severe lymphadenopathy. KICS patients face a higher rate of developing KSHV-related cancers and other malignancies over their lifetime. Due to the cytokine storm induced by vIL-6 secretion in the serum, KICS is accompanied by increased IL-10, hIL-6, and viral loads [57]. vIL-6 drives monocyte proliferation, differentiation into dysfunctional macrophages, and an immune-suppressive phenotype via STAT1/STAT3 activation [10]. It has been suggested that chronic STAT3 activation induced by vIL-6 production may alter the genomic chromatin landscape and enhance inflammatory responses [7]. An increased amount of bromodomain containing 4 (BRD4), a transcription regulator, on chromatin by the prolonged vIL-6 exposure might be responsible for the alteration of chromatin landscape and transcription deregulation observed in KICS [7].

## 7. Summary and Future Direction

The majority of studies on vIL-6 focus on its association with KSHV pathogenesis. Considering the high concentration of vIL-6 in clinical isolates in KSHV-associated tumors and its association with inflammatory signaling, vIL-6 is clearly involved in KSHV-associated disease development. Mouse models indeed proved this notion [65]. An important remaining question is, why does KSHV take the risk of alerting the host immune response by stimulating inflammation? What is the benefit for KSHV to evolutionally maintain vIL-6 homologue, which is not efficiently secreted? The simultaneous activation of NF-κB and STAT3 in non-immune cells triggers a positive feedback loop of NF-κB activation by the hIL-6-STAT3 axis, which is called IL-6 amplification (IL-6 Amp) [135]. We speculate that vIL-6 may be designed to trigger IL-6 Amp in infected/residential cells and utilize the NF-κB and STAT3 signal activation to establish and maintain epigenetically reactivatable KSHV latent chromatins. In this regard, KSHV has captured multiple cellular homologues that are capable of activating NF-kB signaling [136,137,138,139]. In unfortunate circumstances, such as a patient with other chronic inflammation, the localized vIL-6-Amp may become systemic IL-6(s)-Amp, leading to disease development. A previous study also showed that STAT3 colocalized with the LANA/TR complex, suggesting the recruitment of activated STAT3 at the KSHV enhancer (LANA nuclear body) [140,141]. Further studies are needed to understand the biological and virological significance of vIL-6 in KSHV replication cycles.

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
