# Peer review of "The Role of vIL-6 in KSHV-Mediated Immune Evasion and Tumorigenesis"

_viruses, 2024, doi:10.3390/v16121900_

Round 1
Reviewer 1 Report
Comments and Suggestions for Authors
General comment:
This is an ambitious review of a complex and important field by an excellent group working in this area, and the authors have generally done a good job. However, there are a number of minor mistakes (calling MCD multiple Castleman disease, for example), some of which I have pointed out in the specific comments. Also, they sometimes cite minor or controversial papers. I think it needs some more work, and would suggest that the more seasoned authors in the group go though it carefully and give it more of a feel of being written by someone deep in the field. They may also want to highlight their own recent important work on vIL-6.
Specific comments:
1. 36 – I would mention here that vIL-6 is a homolog of human IL-6
2. 39 – MCD is multicentric Castleman disease, not multipl Castleman disease. Also, the role if vIL-6 in a murine model of MCD is ref 44, not 7. And in ref 7, would mention this is a murine model of lymphoma. Also in line 138.
3. 51 section., Suggest expanding to include other factors affecting transcription of vIL-6. For example, there is evidence that X-box binding protein can upregulate vIL-6 expression.
4. 60 – not sure I would mention Notch signaling here.
5. 84 – would not say that vIL-6 is RESPONSIBLE for Rta or viral DNA genes, just that it can influence their expression
6. 135 – They make statement seem for all monoclonal antibodies, but different antibodies may bind toi different sites. Which antibody are they talking about here?
7. Table 1. The cellular origin described is for MCD. I would just remove this bullet.
8. 206 – While KSHV diseases tend to develop in HIV-infected patients, very few cells in KSHV tumors are co-infected by HIV. This section makes it sound like there is frequent co-infection and should be modified. Also, the role of extracellular Tat remains somewhat controversial.
9. 322 – Prognosis of MCD with current treatment is now quite good.
10. 330 Why would vIL-6 be a therapy for MCD??? Related to this, antibody to the IL-6 receptor has not been particularly effective in MCD, and it is not likely that development of a a monoclonal antibody to vIL-6 would be commercially viable. Also, should cite paper here (reference 44) showing that vIL-6 induces hIL-6 in pathogenesis of MCD.
11. General comment – there is a body of literature that vIL-6 binds to receptors in the ER and that mush of its activity is from this binding. This is not really flushed out in this review, and I suggest that it be highlighted more.
Author Response
Please see the attachment. The attached file is the manuscript with highlighted changes to make the follow-up easier. At the end of the file, we included the responses to the reviewer's comments.
Thank you.

Reviewer 2 Report
Comments and Suggestions for Authors
Komaki and Inagaki et al provide an important summary and resource of the numerous disparate studies of the key pathogenic KSHV factor vIL6. The work is appropriate in scope and content. Though the review is nicely organized, its impact could be greatly improved by highlighting in each section the current unknowns and questions. I have included some of these questions below along with some minor corrections and concerns.
1) Two key overarching questions that are relevant for almost all sections are the split between vIL6 function in latent vs lytic cells and whether vIL6 is functioning as a secreted protein via uninfected cells or in a cell-autonomous manner via infected cells (or alternatively, signaling from lytic to latent cells). Where appropriate (such as ln 84 and section 2 and 5i), please comment on where these problems have been resolved and where they remain open (and where they might be very difficult to resolve given the available models).
2) Though I believe this information is present, a separate section or subsection on the similarities, differences, and open questions in secretion of hIL6 and vIL6 would be helpful.
3) Please further discuss the phenomenon of low level vIL6 expressing during latency. Is it known whether this is due to rare populations of spontaneous reactivation or reactivation-independent interferon induced vIL6?
4) Section 4iii. Work on interaction with HIV Tat needs more context here. Are these studies performed in co-infected cells? Or upon exogenous addition/expression of Tat?
5) Section 3i. Is there any connection between the expanded cell tropism of vIL6 and the phenotypes observed in vIL-6 transgenic mice?
6) Ln 67 gp130 is mentioned but not explained till ln 104. Gp80 here is introduced without naming it as IL6R
7) Ln 218/225 Section headers appear misformatted.
8) Ln 263 Typo? C-C Motif chemokine[s]?
9) Ln 331: Target or agent?
10) Consider presenting vIL6’s evolutionary origins and absence/presence in other herpesviruses (or other viral cytokines) in the introduction.
Author Response
Please see the attachment; the responses are at the end of the manuscript.
We also highlighted the changes for better follow-up.
Thank you

Round 2
Reviewer 1 Report
Comments and Suggestions for Authors
The manuscript is substantially improved from the initial submission and the problems have been largely corrected. I do have one minor suggestion; on line 326, suggest that they note that PEL cells express antigens that are found on plasma cells.
Author Response
Thank you very much for your suggestion. We added the following sentence in line 326. "PEL cells express markers that exhibit characteristics of both plasma cells and immunoblasts, corresponding to an intermediate stage in B-cell development between these two cell types."
Please see the attached file with the highlighted sentence.

Reviewer 2 Report
Comments and Suggestions for Authors
The authors have satisfactorily responded to my comments. One note is an error has been introduced in the new text (ln 41) where HSV1/2 is identified as a gamma herpesvirus.
Author Response
I am sorry for the mistake. Thank you very much for pointing this out. We corrected the sentence. Please see the attachment highlighting the corrected sentence.
